# Highly Sensitive Colorimetric Assay of Cortisol Using Cortisol Antibody and Aptamer Sandwich Assay

**DOI:** 10.3390/bios11050163

**Published:** 2021-05-20

**Authors:** Yoonjae Kim, Jongmin Yang, Hyeyeon Hur, Seungju Oh, Hyun Ho Lee

**Affiliations:** Department of Chemical Engineering, Myongji University, Yongin-si 17058, Gyeonggi-do, Korea; yse459@gmail.com (Y.K.); yangjm94@gmail.com (J.Y.); hhuh9801@gmail.com (H.H.); sjtleo55@gmail.com (S.O.)

**Keywords:** cortisol, c-Mab, c-SBA, antibody aptamer sandwich, colorimetric assay

## Abstract

In this study, cortisol, which is a key stress hormone, could be detected sensitively via the colorimetric assay of a polycarbonate (PC) and glass substrate by the sandwich assay of cortisol monoclonal antibody (c-Mab) and cortisol specific binding aptamer (c-SBA). A highly sensitive change in colorimetry with a limit of detection (LOD) of cortisol of 100 fM could be attained on the optically transparent substrate using the antibody aptamer sandwich (AAS) assay by corresponding stacks of 5 nm gold nanoparticles (Au NPs). The Au NPs were conjugated by the c-SBA and the c-Mab was tethered on the PC and glass substrates. For the AAS method, a simple UV-Vis spectrophotometer was adopted to quantify the cortisol concentrations at an absorbance wavelength of 520 nm. Therefore, this study demonstrates the versatility of the AAS method to measure very low concentrations of cortisol in diagnostic applications.

## 1. Introduction

The colorimetric assay is one of the simplest detection methods that has been widely used in the biosensor field [1,2,3]. It has provided many advantages in testing the presence of enzymes, antibodies, and, particularly, low-molecular-weight analytes. In fact, the enzyme-linked immunosorbent assay (ELISA) is one of the common colorimetric assays that qualitatively and quantitatively analyzes the specific target molecules. However, the ELISA has inconveniences in detection time due to the peroxidase’s reaction to induce the color representation or display [1,2,3].

Meanwhile, biosensor systems for neurotransmitter hormone detection have drawn considerable attention due to their available roles toward diagnostic tools for recently increasing psychological disorders. Especially for social mental safety, cortisol has been known as a major biomarker of human psychological stress level, which has frequently been related especially with post-traumatic stress disorder (PTSD), Addison’s disease, Cushing syndrome, and many more [4,5,6]. As an acute response to the adrenocorticotropic hormone (ACTH), cortisol is produced by the adrenal cortex of mammals, whose detection locations have been diversely available in noninvasive sources particularly with tears and saliva [4,5,6,7,8,9]. Depending on the biofluids, the cortisol concentration has been reported to have a wide range of 1 nM. Apart from traditionally developed cortisol assays using fluorescent ELISA [6], new detection protocols of cortisol have been continuously reported using surface plasmon resonance (SPR) sensors, chemiresistors, etc. [7,8,9]. For SPR detection using a simple UV-Vis spectrophotometer instrument, the cortisol measurement method using direct or indirect localized surface plasmon spectroscopy (LSPR)-based platforms has been reported in both a PBS solution and serum at concentrations ranging from 1 to 10,000 ng/mL (2.763 × 10^3^ nM) [5,6,8]. In addition, a chemiresistor sensor using a reduced graphene oxide (rGO) channel could represent a limit of detection (LOD) of 10 pg/mL (27.6 pM) [7]. Very recently, AlGaN-GaN high-electron-mobility transistors (HEMT) have been developed to achieve a LOD of 1 pM [9].

Antibody-based sandwich ELISA and Western blotting have been widely used for detecting small biomarkers [10]. However, the typical ELISA requires a certain amount of time to stain the analyte wells by enzymatic reaction. Moreover, the production of monoclonal antibodies (mAb) is time-consuming, costly, and shows fluctuations in reproducibility [10,11]. Therefore, an alternative strategy of color indexing objects is required for ensuring a robust diagnosis.

For the last two decades, aptamers have been used as exclusive capturing reagents for the sensing of a specific biomolecular target. Aptamers are generally screened through in vitro selection processes, such as the systematic evolution of ligands by exponential enrichment (SELEX). Aptamers or capture probe DNA oligonucleotides can be isolated for a wide range of targets, including low-molecular-weight analytes where they are often called “aptasensors” [11,12,13].

Thus far, aptamers have been selected for a specific assay targeted to cortisol [14,15,16]. Recent developments in aptasensors have employed the combinatorial binding of antibodies and aptamers toward a certain target biomolecule. Basically, sandwich bindings can confirm the binding of the analyte to enhance its sensitivity and stability [15,16].

As the target molecule, it has been frequently reported that cortisol (MW. 362.42) was chosen toward cortisol monoclonal antibody (c-Mab), which was immobilized on a glass substrate as a primary recognition probe, and cortisol specific binding aptamer (c-SBA)-conjugated Au NPs as a secondary recognition probe was adopted as a color indexing agent. It provides a colorimetric detection protocol that is cost-effective compared to fluorescence or radioactivity-based assays because no secondary enzymatic reaction is involved [17,18].

Recently, a colorimetric assay for cortisol detection has been developed in the form of a lateral flow assay (LFA) using 40 nm Au NPs and a cortisol binding aptamer [15]. However, their principle was based on the detachment of the aptamer from the Au NPs upon the binding of cortisol and the sequential aggregation of naked Au NPs accomplishing the LOD at a cortisol concentration of 1 nM [16].

In this study, we developed colorimetric sensing methods using the antibody aptamer sandwich (AAS) assay to detect cortisol on polycarbonate (PC) and glass substrates that are simply measurable in a UV-Vis spectrophotometer, which are easily accessible and inexpensive compared with the conventional ELISA. The PC substrate was UV/ozone (UVO)-activated to form carboxylic groups (-COOH) on the surface, and the glass surface was treated with hydrogen peroxide to generate hydroxyl groups (-OH). Contact angle analysis was performed to investigate the degree of surface modification of the glass substrate. After applying cortisol samples in different concentrations from 1.0 fM to 1.0 μM, the aptamer conjugated with 5 nm Au NPs (8.38 μM) was assembled with cortisol. In order to confirm the presence of bioconjugation between the aptamer and the 5 nm Au NPs, electrophoresis analysis was used using 1.5% agarose gel. At this time, the change in colorimetric absorbance at a wavelength of 520 nm and the relevant shift were examined through UV-Vis absorbance measurement, which can be available in a diagnostic lab.

## 2. Materials and Methods

### 2.1. Preparation of Solution Used for Experiment

For cortisol monoclonal antibody (c-Mab, Abcam Co., Cambridge, UK), 2 mg/mL of solution was diluted to 10 μg/mL. Next, diluted cortisol solutions (Cerilliant Co., Round Rock, TX, USA, 1 mg/mL in methanol) of 1.0 fM, 1.0 pM, 1.0 nM, and 1.0 μM were probed to verify the change in absorbance of the UV-Vis spectrophotometer. Hydrogen peroxide (Deoksan Co., Haman, Korea) was used for the glass surface modification, and (3-glycidyloxypropyl) trimethoxysilane (GPTMS, Sigma Aldrich Co., St. Louis, MO, USA) was dissolved in ethanol. The 5 nm Au NPs (BBI solutions Co., Cardiff, UK) was mixed with an aptamer solution diluted to 10 μM in a volume ratio of 29:1. The aptamer (Bioneer Co., Daejeon, Korea) was used as a solution in which 10 mg of solid powder was diluted to 10 μM in deionized (DI) water. The sequence of these DNA is SH(thiol)-5′-ATG GGC AAT GCG GGG TGG AGA ATG GTT GCC GCA CTT CGG C-3′ [14].

### 2.2. Fabrication of AAS Chip on PC and Glass Substrate

As the first step of PC surface modification, 30 min of exposure under UVO was performed. In order to couple the c-Mab to the PC surface after UVO modification, 2 mL of 0.2 mM of 1-ethyl-3-(3-dimethylaminopropyl)-carbodiimide (EDC) in DI water was treated by gently dip-coating for 30 min; then, 2 mL of 0.3 mM of *N*-hydroxysuccinimide (NHS) was also treated for 30 min as chemical activators, as shown in Figure 1.

To modify the glass surface, 30–35% hydrogen peroxide (H_2_O_2_) treatment was used. The antibody aptamer sandwich (AAS) assay was solely used using the colorimetric absorbance of 5 nm Au NPs conjugated with the c-SBA. Then, 5% *v*/*v* (3-glycidyloxypropyl) trimethoxysilane (GPTMS) in ethanol was used to confer glycidyl functionality to the surface, and then to bind with cortisol monoclonal antibody (c-Mab) through epoxy-amine bond formation with the amine groups of c-Mab.

After capturing and recognizing cortisol through the antigen–antibody reaction for 30 min, cortisol binding (30 min) aptamer-conjugated 5 nm Au NPs were combined to realize the sandwich assay. After secondary binding of the c-SBA-conjugated Au NPs, DI water rinsing was carried out.

### 2.3. Characterization of Glass and PC Substrates

XPS (X-ray photoelectron spectroscopy: ESCA2000, VG Microtech, England) showed elemental components of UVO-modified PC. The XPS analysis out of C 1 s binding energy regions was deconvoluted to identify corresponding peaks. The contact angles were measured by an automatic contact angle analyzer (Pheonix 300 Touch, SEO, Suwon, Korea) with 3 μL of deionized water droplets.

### 2.4. Characterization of Aptamer-Conjugated Au NPs

The Au NPs were conjugated with c-SBA by room temperature incubation. The conjugated Au NPs were recovered by centrifugation with a microcentrifuge tube with filtering (MWCO 10,000, vivaspin 500, Cole Palmer, USA). To remove unbound or noncoupled c-Mab and c-SBA, the PC and glass surfaces were rinsed by DI water for every step. The aptamer-conjugated Au NPs were analyzed by FTIR (FT/IR460, JASCO, Japan) in attenuated total reflection (ATR) mode. Agarose gel (1.5%) was made by dissolving 0.45 g of agarose in 30 mL of 1× SB (sodium borate) buffer. In an electrophoresis system (Biorad Co., Richmond, CA, USA), 100 V was applied at both Pt wire electrodes and 10 μL of gold, and gold solutions were introduced into each sample well of agarose gel [19].

### 2.5. Antibody Aptamer Sandwich Assay Procedure

Different concentrations of cortisol standard solution (1.0 fM~1.0 μM) were applied to the c-Mab surface for 30 min as a primary recognition or capturing, and the surface was rinsed by DI water sequentially. After air drying, 0.1 mL of the c-SBA 5 nm Au NPs (8.38 μM) solution was applied for secondary binding toward the primarily captured cortisol.

## 3. Results

It is essential to identify the optimum c-SBA concentration applied to cortisol captured surface. We measured the colorimetric AAS absorbance by changing the concentration of standard sample cortisol from 1.0 fM to 1.0 μM.

Figure 1 shows a schematic illustration of the procedures of the AAS assay on the PC and glass substrates. For the PC substrate, the surface was UVO-treated first; then, it was chemically surface-modified with EDC/NHS. Primary binding of cortisol on c-Mab was carried out, and aptamer binding for sensing and color representation was then performed.

For the glass surface, the glass substrate was immersed in hydrogen peroxide solution for the formation of hydroxyl functional groups (-OH). After oxidation processing, GPTMS was incubated on glass substrates for 30 min so the terminal functional group could possess an epoxy group. Here, the surface was modified so that the amine group of c-Mab coupled well, which selectively binds on cortisol. The c-Mab was immune-bound with cortisol samples, and the cortisol was then doubly bound to aptamer-conjugated 5 nm Au NPs.

Figure 2a shows XPS spectra results of the UVO-treated PC surface to confirm the generation of the carboxylic group, which can be basic for the surface chemistry of protein conjugation.

In addition, to confirm the extent of the surface modification of glass, contact angle analysis for each stage of surface modification was performed, as shown in Figure 2b.

Figure 3a shows the FTIR spectra of c-SBA-conjugated Au NPs on the IR sampling card.

In order to confirm the conjugation between 5 nm Au NPs and aptamers used for the secondary cortisol recognition and capture, agarose gel electrophoresis was performed, which can be seen in Figure 3b.

Figure 4a shows photographic images of the PC colorimetric assays, which were captured after six months of the assay with the samples of the control, 1.0 pM, 10 pM, 100 pM, and 1.0 nM cortisol concentrations.

Figure 4b shows photographic images of glass colorimetric assays of the control, 1.0 fM, 1.0 pM, 1.0 nM, and 1.0 μM cortisol samples. After cortisol was dually bound by the c-Mab and cortisol-specific binding aptamer-conjugated 5 nm Au NPs, a reddish surface could be clearly demonstrated in Figure 4b.

Figure 5a shows the extinction spectra with various concentrations of cortisol solutions at 1.0 fM, 10 fM, 100 fM, and 1.0 pM on the PC AAS sensor.

Figure 5b shows the extinction spectra with various concentrations of cortisol solutions at 1.0 pM, 1.0 nM, and 1.0 μM on the glass AAS sensor.

## 4. Discussion

For the design of the AAS assay method, there can be an alternative strategy to conjugate the aptamer as a primary recognition element on the PC and glass substrates, unlike this study. However, there were two reasons to adopt the antibody as the primary recognition part, as shown in Figure 1. First, it was not efficient to tether the aptamer, even though the amine (-NH_2_)-functionalized aptamer at the 5′ end was used, to the PC substrate and glass substrates. Especially for the PC substrate, the yield of carboxylic group formation was extremely sensitive for the aptamer coupling through the EDC/NHS chemistry, and, as a consequence, the efficiency of the aptamer immobilization was significantly lower than that of the antibody immobilization (data not shown). Second, in terms of stability, the aptamer could be more vulnerable to decompose under various enzymes (nucleases and proteases) of the human fluid sample such as saliva.

Based on the C 1s XPS spectra in Figure 2a, the carbon constituent of the PC surface was clearly oxidized to form carboxylic groups. The three C 1s peaks of the pristine PC in Figure 2a represent hydrocarbon (-CH_2_- at 284.6 eV), a ketone or ether group (-C-O-C- at 286.8 eV), and a carbonate group (-O-(C=O)-O- at 290.6 eV). Meanwhile, the UVO-modified PC surface could show a high carboxylic group content (-(C=O)-O- at 288.8 eV). With reactive species of oxygen radicals under the UVO environment, carbon from hydrocarbons and carbonate bonds are oxidized to form ketone and carboxylic carbon [20]. Therefore, the UVO treatments could add a large amount of oxidized carbon groups to the surface.

The contact angle of the untreated glass substrate with water was 60.29°, and when both hydrogen peroxide and GPTMS were treated consecutively, they were 29.87° and 16.17°, as shown in Figure 2b, respectively [20]. It can be seen that the glass substrate was inter-converted from a hydrophobic state to a hydrophilic state depending on the surface modification chemistry. For example, the epoxy group after GTPMS treatment could clearly show a highly hydrophilic surface, which was subjected to a labile substitution reaction with the amine group of c-Mab. Therefore, the applied cortisol samples to the c-Mab-conjugated glass surface could be varied from 1.0 fM to 1.0 μM to be immune-bound to the c-Mab.

In Figure 3a, peaks around 2960 cm^−1^ could correspond to the thiol group of the c-SBA on Au NPs. Multiple and broad peaks around 3400 cm^−1^ are believed to be from hydrogen bonds of complementary hybridization within oligo-DNA of c-SBA [21]. The thiol group peak and hydrogen bond peaks are apparent only for the aptamer-conjugated Au NPs. Meanwhile, the peaks at 1150 and 1215 cm^−1^ are from phosphodiester backbones of DNA. The peak at 1450 cm^−1^ corresponds to base-sugar vibrations and the peak at 1650 cm^−1^ shows a weak stretching vibration of the DNA bases. The peaks for phosphodiester and base-sugar are clearly detected exclusively for SAMs of probe DNA. In addition, peaks at 800 and 930 cm^−1^ are also observed as glycyl or the functional group of GPTMS, respectively [21].

When the two lanes of electrophoresis analysis were compared in Figure 3b, the 5 nm Au NPs’ band of lane (1) could be seen to run farther from the baseline than that of lane (2). As no remaining Au NPs were identified in the well of lane (1), it clearly shows that the conjugation between the aptamer and the Au NPs occurred successfully. In addition, the 5 nm Au NPs were adopted in this study to induce the more efficient conjugation of c-SBA rather than larger sized Au NPs. Moreover, it was less accessible to aggregation to show the extra colorimetric change with the 5 nm Au NPs.

In Figure 4a, the images were obtained with the PC substrate AAS sensors six months after the colorimetric measurement. As shown in Figure 4a, reddish colorimetric spots were still detectable for the 100 pM and 1.0 nM samples. However, the control, 1.0 pM, and 10 pM samples showed completely purple-colored spots.

In Figure 4b, it was found that the reddish colors after N_2_ purge drying for all samples indicated no severe aggregation of 5 nm Au NPs. This could originate from an advantage of the glass substrate. However, there were no recognizable differences among samples with the naked eye. As a consequence, a sensitive UV-Vis measurement was inevitable. In addition, a long time preservation of color marks was not available with the glass substrate.

As shown in Figure 5a, the control, 1.0 fM, and 10 fM samples could show small peaks at the 520 nm absorbance peak. However, 100 fM and 1.0 pM samples could show higher absorbance and shifted peaks from the 520 nm [1]. From the data in Figure 5a, the limit of detection (LOD) of cortisol could be defined as 100 fM. In addition, there were no distinct aggregations that were frequently detected as a shoulder peak in the UV-Vis extinction spectra around 700~800 nm.

In Figure 4b, the unique red color of Au NPs did not show a significant difference under bright illumination when viewed with the naked eye. However, the colorimetric change according to the cortisol concentration was obviously confirmed in the accurate measurement of extinction spectra using a UV-Vis spectrophotometer in Figure 5b. The higher the concentration compared to the control sample, the higher the absorbance peak in Figure 5b. Compared to the control test, as the concentration of cortisol increased, it was confirmed that the absorbance at a wavelength around 560 nm increased proportionally, which shows a clear linearity between cortisol concentration and the magnitude of absorbance. The maximum peak positions (562, 560, and 556 nm) evolved depending on the cortisol concentrations of 1.0 pM, 1.0 nM, and 1.0 μM.

## 5. Conclusions

A novel highly sensitive cortisol assay based on the colorimetric method was developed using the AAS method. PC and glass substrates were functionalized to conjugate c-Mab, and it was immune-bound by cortisol, which was sequentially bound by the cortisol-specific aptamer-capped 5 nm Au NPs. The colorimetric change in UV-Vis spectra in the 520 nm absorbance achieved an LOD as low as 100 fM of the cortisol sample without the severe aggregation of 5 nm Au NPs.

## Figures and Tables

**Figure 1 biosensors-11-00163-f001:**
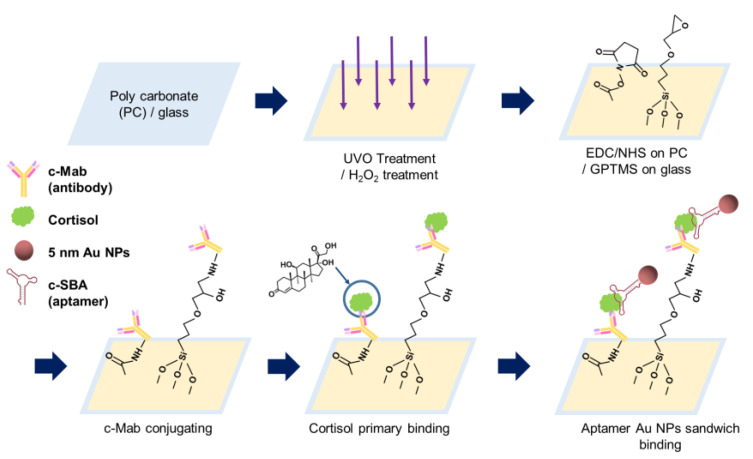
Schematic process diagram of cortisol antibody and aptamer sandwich assay on PC and glass substrate.

**Figure 2 biosensors-11-00163-f002:**
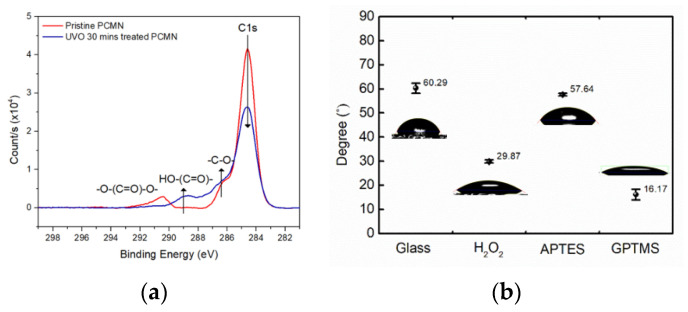
(**a**) XPS spectra of C 1s of pristine PC and UVO-treated PC; (**b**) contact angle analysis depending on surface-modified states of glass.

**Figure 3 biosensors-11-00163-f003:**
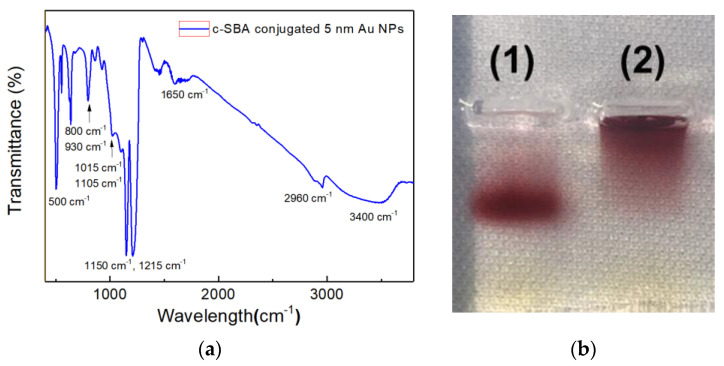
(**a**) FTIR spectra of c-SBA-conjugated 5 nm Au NPs; (**b**) 1.5% agarose gel electrophoresis image of (1) aptamer-conjugated 5 nm Au NPs and (2) pristine 5 nm Au NPs.

**Figure 4 biosensors-11-00163-f004:**
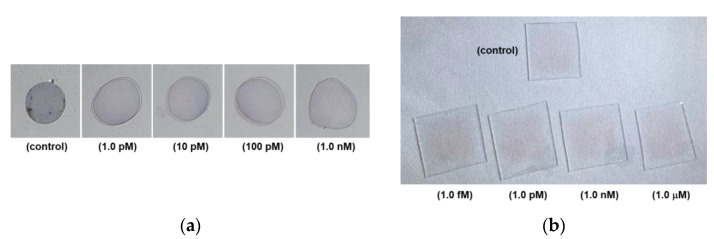
(**a**) Photograph images of PC colorimetric assay taken after six months: control, 1.0 pM, 10 pM, 100 pM, and 1.0 nM; (**b**) photographic images of glass colorimetric assay: 1.0 pM, 1.0 nM, and 1.0 μM.

**Figure 5 biosensors-11-00163-f005:**
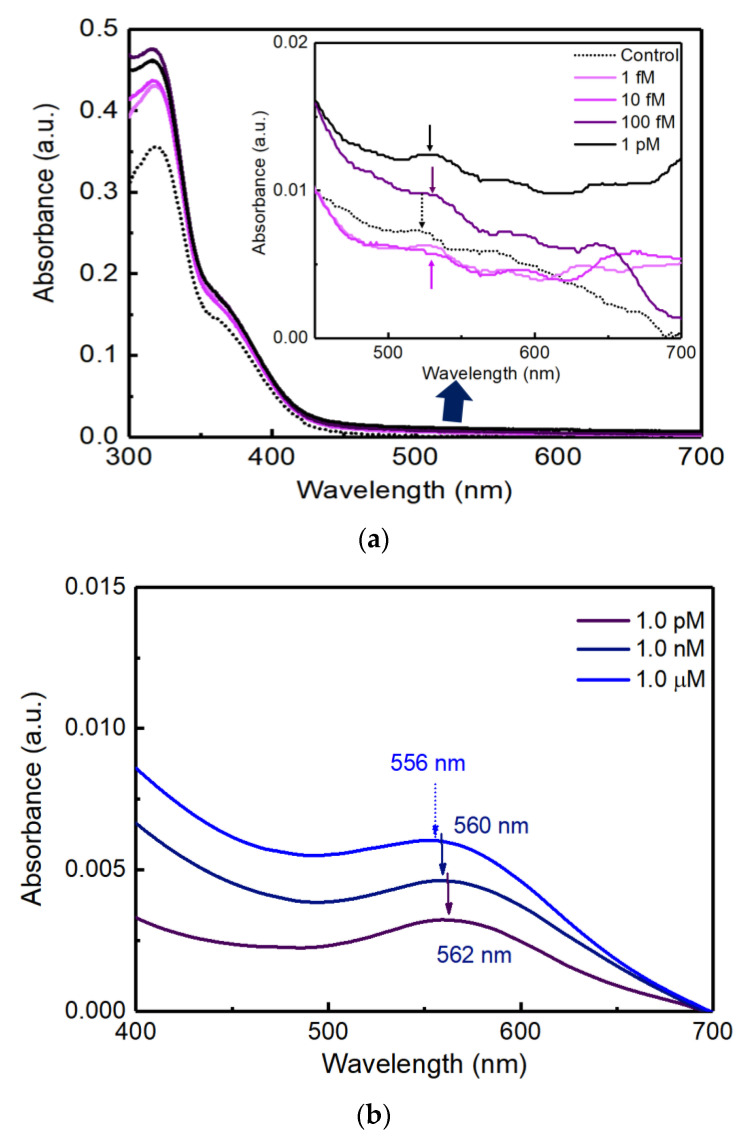
(**a**) Extinction spectra of UV-Vis depending on cortisol concentration from 1.0 fM to 1.0 pM on PC substrate. (**b**) Extinction spectra of UV-Vis measurement depending on cortisol concentration from 1.0 pM to 1 μM.

## Data Availability

Not Applicable.

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
