# Peer review of "Highly Sensitive Colorimetric Assay of Cortisol Using Cortisol Antibody and Aptamer Sandwich Assay"

_biosensors, 2021, doi:10.3390/bios11050163_

Round 1
Reviewer 1 Report
This is a very interesting paper. There is a great body of evidence. The manuscript is well-written, and the results are well-presented. The following minor comments should be addressed by the authors.
- It is not clear whether the authors aimed to develop this novel test to be used in everyday routine of hospitals and diagnostic labs or research labs only or both. Please explain.
- Why did the authors choose cortisol and not another hormone or metabolite with diagnostic potential?
Author Response
- Reviewer #1: This is a very interesting paper. There is a great body of evidence. The manuscript is well-written, and the results are well-presented. The following minor comments should be addressed by the authors.
Response: Thank you very much for kind decision of revision.
- It is not clear whether the authors aimed to develop this novel test to be used in everyday routine of hospitals and diagnostic labs or research labs only or both. Please explain.
Response: Thank you very much for kind comment. The method can be used in diagnostic lab since the analysis was based on UV-Vis spectra measurement. However, if more drastic color change is further developed, this scheme is available for point-of-care test (POCT).
So, authors revised the manuscript in page 3, line 103,
“At this time, the change in colorimetric absorbance at wavelength 520 nm and relevant shift were examined through UV-Vis absorbance measurement, which can be available in diagnostic lab.”
- Why did the authors choose cortisol and not another hormone or metabolite with diagnostic potential?
Response: Thank you very much for sincere comment on this paper. Basically, first, the amount of cortisol in saliva is relatively higher than other hormones (e.g. dopamine, serotonine) ranging from 0.6 ng/mL to 10.4 ng/mL, which is rationally detectable with simple scheme [4-8,15,16]. Second, unlike other hormones, there are more diseases directly related to the level of cortisol than that of other hormones.
So, in page 1, authors revised the manuscript,
“Especially for the social mental safety, cortisol has been known as a major biomarker of human psychological stress level, which has been frequently related especially with post-traumatic stress disorder (PTSD), Addison’s disease, Cushing syndrome, and many more [4-6].”

Reviewer 2 Report
This work presents a colorimetric assay for the detection of cortisol. This immunoassay is superior to the conventional enzymatic colorimetric method in terms of effectiveness. I think this paper is innovative enough for Biosensors and will attract to a large audience. Therefore, I recommend that this paper be accepted for publication after a major revision. The specific comments are as follows.
- The abstract expounds the main contents of the research work, but lacks the summary of significance and prospect of this work
- The introduction section seems tedious, the authors could cut some unimportant words. I think the author should select some representative high-quality works for careful analysis.
- The arrow indication in Figure 1 is not clear. After the third step it should be indicated with an arrow to the far left of the next line.
- The experimental section does not seem to contain information about contact angle testing.
Author Response
- Reviewer #2: This work presents a colorimetric assay for the detection of cortisol. This immunoassay is superior to the conventional enzymatic colorimetric method in terms of effectiveness. I think this paper is innovative enough for Biosensors and will attract to a large audience. Therefore, I recommend that this paper be accepted for publication after a major revision. The specific comments are as follows.
Response: Thank you very much for kind decision of revision.
- The abstract expounds the main contents of the research work, but lacks the summary of significance and prospect of this work
Response: Thank you very much for kind comment on deficiency on abstract part.
So, authors revised the manuscript, in the abstract,
“In this study, cortisol, which is a key stress hormones, could be detected sensitively through on colorimetric assay on polycarbonate (PC) and glass sub-strate by sandwich assay of cortisol monoclonal antibody (c-Mab) and cortisol specific binding aptamer (c-SBA). Highly sensitive colorimetric change having a limit of detection (LOD) of 100 fM cortisol could be attained on the optically transparent substrate using the antibody aptamer sandwich (AAS) assay by corresponding stacks of 5 nm gold nanoparticles (Au NPs). The Au NPs were conjugated by the c-SBA and the c-Mab was tethered on the PC and glass substrates, respectively. For the AAS method, a simple UV-Vis spectrophotometer was adopted to quantify the cortisol concentrations at 520 nm wave-length absorbance. Therefore, this study can open up more versatility of the AAS method to measure a very low concentration of cortisol in a diagnostic application purpose.”
- The introduction section seems tedious, the authors could cut some unimportant words. I think the author should select some representative high-quality works for careful analysis.
Response: Thank you very much for the unnecessary parts in the Introduction.
So, authors removed many tedious and repetitive parts as follows:
At page 1, line 30, “between antibody and color producing enzymes such as peroxidase.”
At page 1, line 42,43,44, “The cortisol assay has been”, “enzyme-linked immunosorbent assays”, “In addtion”
At page 2, line 52, “In this thread, the detection of the cortisol has been one of intensively investigated topics in recent publications including electrochemical and electrical detection princi-ples [4-9].”
At page 2, line 59, “Furthermore, production and preservation of peroxidase for the ELISA is somewhat limited due to instability of the enzyme.”
At page 2, line 62, “oligonucleotide-based recognition elements, referred to as”
At page 2, line 65, “which enables the isolation of nucleic acid aptamers from combinatorial libraries.”
At page 2, line 68, “proteins, antibiotics, viruses, and organic/inorganic compounds, and mainly used in analytical and biosensor applications”
At page 2, line 75,
“In general, glass does not have enough hydroxyl group and cannot be chemically con-jugated with antibodies. However, their optical clearance can be always guaranteed as sensor substrate especially for ultrasensitive colorimetric assay.”
At page 2, line 89,
“The colorimetric method was based on two different-typed aptamers and detachment of aptamer from the 40 nm Au NPs by high ionic strength solution.”
- The arrow indication in Figure 1 is not clear. After the third step it should be indicated with an arrow to the far left of the next line.
Response: Thank you very much for kind correctional comment. The Figure 1 was replaced and the manuscript was revised.
- The experimental section does not seem to contain information about contact angle testing.
Response: Thank you very much for kind check about the missing part. Authors add the experimental details as follows.
At page 3, line 137,
“The contact angles were measured by an automatic contact angle analyzer (Pheonix 300 Touch, SEO, Korea) with 3 mL of deionized water droplet.”

Round 2
Reviewer 2 Report
The revised version can be accepted for publication.